# Investigating the Impact of Smart Tourism Technologies on Tourists' Experiences

Yuwen Zhang, Marios Sotiriadis  and Shiwei Shen *

Ningbo University—University of Angers Joint Institute, Ningbo University, Ningbo 315211, China;
zhangyuwen05291124@163.com (Y.Z.); sotermarios@outlook.com (M.S.)
* Correspondence: shiwei_shen@163.com; Tel.: +86-13957870550

**Abstract:** The adoption and implementation of smart technologies in tourism destinations and visitor attractions to enrich tourists' experiences and improve their satisfaction has become a new trend. The main purpose of this study was to explore the influence of the dimensions/attributes of smart technologies on tourism experience in the context of visitor attractions and related outcomes (satisfaction and post-consumption behavioral intentions). The Liangzhu Museum, Zhejiang Province, China, was used as the study area, and data were collected from 486 visitors and analyzed with a regression model. The results show that accessibility and interactivity affect smart technology-enhanced experiences. Tourists' perceived value of smart technologies is significantly related to their satisfaction. Smart technology positively impacts word-of-mouth recommendations, revisit intention, and willingness to pay a price premium. Therefore, visitor attractions could improve tourists' experiences by designing better infrastructure and services that incorporate the key dimensions of smart technologies, which would also improve their competitiveness.

**Keywords:** smart tourism technologies; attributes; consumer behavior; tourism experience; visitor attractions; China

## 1. Introduction

With the explosive growth of the userbase of the global mobile Internet and smart devices [1], technology is no longer an auxiliary tool and the use of smart technology has become an inevitable requirement for industrial development [2]. The integration of tourism and smart technologies is logical against this backdrop. Smart technology helps tourism destinations improve the management efficiency of tourism resources, promote the maximum utilization and sustainable development of tourism resources, and improve the quality of life of permanent residents and tourists.

An increasing number of visitor attractions have adopted smart technologies, such as artificial intelligence (AI), cloud computing, and Internet of Things (IoT), to enrich the tourism experience. For example, virtual reality (VR) technology provides tourists with a physically, spiritually, and emotionally integrated tourism experience [3]. Tourism destinations become 'smart' by implementing smart technology to increase competitiveness [4]. Tourists use available smart technologies for decision making, such as organizing travel plans on their mobile phones, interacting with other tourists, and sharing their tourism experiences [5].

With the popularization of mobile smart devices, the number of mobile tourism users continues to grow, as does the penetration rate of mobile tourism and mobile Internet users. By the end of 2018, the number of mobile tourism users had grown to 620 million, with a penetration rate of 42.6 percent [6]. In order to support the rapid growth of mobile tourism users, in November 2020 the Chinese government introduced a plan for smart upgrading through smart tourism technology [7]. According to China Investment Consulting Network statistics, China's AAAAA level visitor attractions have invested an average of more

than CNY 10 million in smart infrastructure. The funds are mostly used to install Wi-Fi coverage in visitor attraction areas; produce attraction maps using GPS; and create smart management, mobile payment, and reservation platforms [8].

The phenomenon of smart tourism has been analyzed from various perspectives. One interesting study was conducted by Križaj et al. (2021) [9] that took a supplier/destination perspective with an operational and innovation adoption-based approach and aimed to investigate the technological content of smart tourism projects implemented in Europe according to specific criteria, i.e., a smart actionable classification model (SACM). Their study contributed to a better understanding of the smartness paradigm, as the term "smart" is frequently used as a fashionable buzzword for "smartwashing". It was found that the vast majority of projects designed as "smart" mainly pursue environmental sustainability goals but do not meet specific digital technology criteria or social sustainability objectives.

Our study takes a demand/tourist perspective and focuses on the technology-enhanced experience. It contributes to a better understanding of the actual content and operational implementation of smart tourism technologies (STTs) by exploring the specific attributes of these digital technologies in the context of visitor attractions. Therefore, our study is not about "smartwashing" but about smartness implementation at the micro/business level (visitor attractions) with the aim of enhancing the tourism experience and increasing visitor satisfaction, as well as the effectiveness of resource management.

The extant research indicates that smart tourism technologies (STTs) are related to tourism experiences [10–12] as a significant influencing factor determining the satisfaction level of tourists [13,14]. Therefore, many tourism destinations and attractions have adopted and implemented STTs to provide tourists with a convenient, friendly, and personalized tourism experience to increase their satisfaction. Research in this area mainly comprises case studies on smart tourism destinations [15,16] or the application of STTs in tourism destinations or tourist attractions [17,18]. A few studies have explored the relationship between STT attributes, tourism experiences, and tourist consumption behavior. This is the first knowledge gap that our study attempts to address.

Buhalis suggested the conceptualization of STT-related experiences encompassing four dimensions/attributes: information, interactivity, accessibility, and personalization. Some scholars recently proposed a fifth dimension, security [19,20]. It is believed that the manner in which and extent to which destinations and attractions implement the five STT dimensions/attributes affects tourists' perception of the service experience of STTs, as well as their consumption behavior. Our research question was: "what is the impact of STT attributes on tourists' satisfaction and their behavioral intentions within the setting of visitor attractions?".

In the present study, we addressed the above research question by investigating the causal relationship between the attributes of STTs and tourists' behavioral intentions through the mediating role of tourist satisfaction. The chosen tourism attraction was Liangzhu Museum, China, which is regarded as a typical example of Chinese museums that have implemented a smart infrastructure and management system using some of the latest STTs. Data were analyzed with a regression model.

The expected contribution of this study is threefold. It contributes to the understanding of tourists' perceptions of STT attributes, which may help to enrich research on STT attribute theory. Secondly, the research model established in this study helps us to understand the relationship between STT attributes, tourist satisfaction, and behavioral intention. Thirdly, the conclusions of this study can help visitor attractions maintain and improve their competitiveness.

## 2. Literature Review

STTs include a series of technologies and services, such as the Internet of Things (IoT), cloud computing, artificial intelligence (AI), mobile communication, radio frequency identification devices (RFID), smart devices, augmented reality (AR), virtual reality (VR), mobile payment, mobile communication, social networking sites, and tourism-related plat-

forms [4,19,21]. The literature indicates that STTs enrich tourists' experiences, satisfaction, and behavioral intention [22].

### 2.1. Attributes of Smart Tourism Technologies

STTs refer to applications that enhance the tourism experiences and generate added value [23]. The extant literature has explored the separate impact of specific STTs on the tourism experience. A study examined the utility of big data in tourism in collecting and analyzing qualitative and quantitative information on demand [24]. Likewise, AI contributes to the elaborate design of products and experiences that conform to consumer preferences based on big data processing [25]. Mobile technology offers tourists more convenient conditions since they can use smartphones, tablets, or other mobile devices to contact any person at any time from anywhere to interact and share experiences [26]. Various online social platforms and social media have become the main places for tourists to share travel-related information and have changed the way tourists share their experiences [27]. Similarly, AR and VR technologies allow tourists to experience interactive computer-supported environments [28,29].

STTs affect tourists' opinions and perceptions and influence their behavioral intention [19]. The study by Buhalis et al. [19] suggested a conceptualization of STT attributes that include four key elements, i.e., information, accessibility, interactivity, and personalization. A fifth attribute—security—was put forward [19,30]. This study argues that Buhalis' conceptualization constitutes a valuable theoretical basis for attaining our research aims.

### 2.1.1. Information

Information is the combination of qualitative, credible, and accurate input and comments generated by tourists about tourism destinations and suppliers/attractions [19]. Kim and Hiemstra pointed out that information quality plays an essential role in tourists' perception about destinations and attractions [31]. Likewise, information reliability is critical at the initial search stage [32]. By utilizing STTs, it is easy for tourists to expand the depth and breadth of relevant tourism information. Such information is very helpful for gathering inspiration, decision-making, and enjoying the experience of visiting an attraction. In summary, the attribute of information is a valuable dimension of STTs and significantly contributes to achieving better efficiency and effectiveness in decision-making by tourists.

### 2.1.2. Accessibility

Tourists obtain and use travel-related information using different types of STTs. Accessibility refers to the degree of difficulty of tourists to access and use tourism information provided by tourism destinations/suppliers through various STTs [30]. High-quality accessibility of STTs facilitates the task for tourists, thereby improving the perceived ease of use of STTs. When STTs are easy to access and use, tourists will relish using these digital technologies to obtain information at all stages of their trips and visits. In doing so, tourists acquire the means to enhance their experience and level of satisfaction [33].

### 2.1.3. Interactivity

Another facilitating factor/attribute is interaction. The interactivity of STTs is defined as the interaction between interested/involved stakeholders [34]. The interactivity of STTs can facilitate timely and active two-way communication between stakeholders [19]. The attribute of interactivity considerably facilitates the task of information searching. This high-level interactivity impels tourists to actively use STTs [35] and provide comments and feedback [36]. The final output of this dimension is a significant and positive impact on the smooth flow of the tourism experience [35].

### 2.1.4. Personalization

Personalized service can meet tourists' requirements for customization and maximize tourists' satisfaction in tourism destinations and attractions [37]. The research by Schaupp and Bélanger [38] revealed that customized services reduce the opportunity cost and duration of information searching and therefore improve tourists' satisfaction [38]. Likewise, Park and Gretzel [39] found that offering personalized services improves the service awareness of tourists [39]. By collecting data on consumer behavior, STTs provide insights and better knowledge on consumers' habits and preferences and then provide them with appropriate and suitable offerings and products [40].

### 2.1.5. Security

Security refers to the degree of confidentiality of private information when engaged in various transactions [39]. In tourism destinations, the extent of use of STTs is determined by the tourists' perception about the respect of privacy and of the personal information shared [30]. When tourists feel that personal information security is under threat/risk, they will not complete the transaction due to concerns of privacy and safety [31,40,41].

Tourists' perception of STTs is an integral part of the tourism experience and an antecedent influencing tourists' satisfaction and behavioral intention [14,42]. Therefore, it is essential to understand tourists' perceptions about STTs. This study evaluates the impact of the five attributes mentioned above of STTs—information, accessibility, interactivity, personalization, and security—on the perceived value of tourists' experiences.

### 2.2. Perceived Value

The study of perceived value started with the customer's perceived value in consumer behavioral science. The focus on tourism perception began in the 1970's, which is also considered important research content in social psychology [43]. Zeithaml first proposed the perceived value theory; she concluded that perceived value is the comprehensive evaluation by consumers after weighing the perceived benefits and perceived costs [44]. Huang concluded that tourists' perceived value is "a comprehensive evaluation of whether the attributes of products and services used in the whole tourism process meet their expectations and meet their demands based on their consumption experience and consumption preferences" [45,46]. In this research area, researchers have studied the dimensions of perceived value and outcomes, i.e., satisfaction and behavioral intentions. Academic research has demonstrated that tourists' perceived value positively influences their satisfaction [47] and is a key factor in determining tourists' purchase choices [46] and in predicting their consumption behavior and revisit intention [48]. Moreover, the concept of perceived value has expanded from two-dimensional (perceived benefits and costs) to multi-dimensional because customer perceived value is a dynamic and varying concept, depending on different situations [45].

In smart tourism, the STTs and the related services constitute a facilitating factor enhancing tourists' experiences. It is therefore necessary to explore tourists' perceptions of the influence of STTs, as a comprehensive set of facilitating digital technologies and not separately, on tourists' experiences [49,50]. This kind of approach is taken for the first time here.

### 2.3. Tourist Satisfaction

The concept of tourist satisfaction originates from the marketing literature. Customer satisfaction is the consumer's judgment of whether products or services meet their requirements and expectations [49]. Tourist satisfaction is considered the result of a comparison between tourists' expectations and the actual tourist experience [50]. The research stream on tourist satisfaction focuses on the influencing factors and mechanisms [51]. It mainly adopts empirical research by examining the way that some factors, such as tourist expectations [52], perceived value [53], price [54], tourism destination image [55], sense of awe [56], and nostalgia [57], affect tourist satisfaction.

In addition, some researchers have proposed that tourist satisfaction is affected by the infrastructure and amenities of tourism attractions [58], service facilities [59], and different contexts [60]. In summary, the extant literature mainly focuses on tourists' satisfaction by using this factor/construct to assess tourists' overall satisfaction. The concept of satisfaction can be divided into specific and overall satisfaction [61–63]. Specific satisfaction refers to the immediate emotional response of tourists when they use certain tourism services or products when traveling. At the same time, the overall satisfaction is a summary psychological state that integrates all the immediate emotional responses after completing the tourism experience [64]. Specific satisfaction affects overall satisfaction. Therefore, even if tourists are not particularly satisfied with a certain service they will still be happy with the whole experience if they consider and assess the entire experience positively. This study argues that the attributes of STTs affect the specific satisfaction of tourists.

Academic research suggests that satisfaction involves and generates positive outcomes, such as positive word-of-mouth and repurchase/revisit intention [65], and equally affects the continuous use of technology [66]. However, various studies have demonstrated that STTs improve satisfaction through the whole travel life cycle (pre-, during, and post-trip). However, it has not been clearly demonstrated whether the influence on behavioral intention originates from satisfaction with the services of tourism suppliers (i.e., hotels, airlines, car rentals, travel agencies) or with the use of various STTs [67]. This study focuses on satisfaction based on tourists' perceptions of the influence of the means/tools (STTs) on the experience, not the services of the tourism suppliers themselves.

*2.4. Post-Experience Behavioral Intention*

Behavioral intention is the subjective probability of a person/consumer performing a specific behavior [68]. The best way to predict consumers' probable behavior is to apprehend consumers' behavioral intentions [69]. This concept has attracted scholars' attention. However, due to the differences in the research background, scholars' classification and measurement of customer behavioral intentions is not homogeneous in the marketing literature. Parasuraman suggested four aspects that could measure behavioral intention: purchase intention, word-of-mouth, price sensitivity, and complaint behavior [70]. Boulding considers two important dimensions of measuring behavioral intention: repurchase intention and word-of-mouth [71]. Hence, the research area of consumption behavior intention mainly focuses on two aspects: repurchase intention [72–75] and word-of-mouth or recommendation intention [76–78].

Gronholdat indicates that the dimensions of behavioral intention should also include willingness to accept price fluctuations [79]. Similarly, Crompton also considers that customer behavior intentions consist of loyalty, repurchase, recommendation, and willingness to pay higher prices [80]. Dong and Jin summarized the different suggestions and concluded that customer behavior intention should be assessed based on three measures/criteria: repurchase intention, recommendation intention, and willingness to pay a premium price [81].

In the tourism literature, Chen defines tourist behavior intention as "a prediction of the possibility of revisiting and recommending destinations to relatives and friends" [82]. Revisit intention and recommendation intention are measured separately as two dimensions in this definition. Nevertheless, a satisfactory tourism experience may not guarantee that tourists will revisit the destination, but it will likely have a good reputation and positive word-of-mouth effect. Therefore, it is more appropriate to consider the tourists' intention to revisit/repurchase and the willingness to recommend separately. Consequently, it is more suitable to distinctly evaluate the revisit intention and recommendation intention.

Dong and Jin [81] also suggested a third criterion/dimension—willingness to pay a premium price—for assessing behavioral intention. This factor measures the willingness of a consumer to pay a high price for a commodity or service and is called willingness to pay a price premium (WPPP). Many scholars have examined the first dimensions (revisit intention and willingness to recommend [83,84]; however, few studies have considered

the third—WPPP—a measurement variable. Therefore, this study uses three measurement dimensions/criteria for tourists' behavioral intentions.

## 3. Research Model and Hypotheses

### 3.1. Research Hypotheses

3.1.1. Attributes of Smart Tourism Technologies

The services provided by STTs in tourism destinations and visitor attractions exert an important influence on tourists' experiences [85]. In the process of experiencing the infrastructure and services provided by STTs, tourists' evaluation of whether STTs meet their expectations and requirements is the tourists' perceived value of STTs. Based on the classification of STTs' attributes [19,30], this research explores the value resulting from and generated by five features—i.e., information, accessibility, interactivity, personalization, and security—as perceived by tourists. Based on this argument, this study postulates the following hypotheses in the setting of visitor attractions:

**H1a.** *The information of STTs has a significant impact on the perceived value of smart technology-supported experience in visitor attractions.*

**H1b.** *The accessibility of STTs significantly impacts the perceived value of smart technology-supported experience in visitor attractions.*

**H1c.** *The interactivity of STTs significantly impacts the perceived value of smart technology-supported experience in visitor attractions.*

**H1d.** *The personalization of STTs significantly impacts the perceived value of smart technology-supported experience in visitor attractions.*

**H1e.** *The security of STTs significantly impacts the perceived value of smart technology-supported experience in visitor attractions.*

3.1.2. Perceived Value of Smart Technology-Enhanced Tourism Experience and Satisfaction

Perceived value is a comprehensive evaluation conducted by tourists based on perceived benefits and costs [44]. Previous studies have shown a strong link between perceived value and satisfaction [86]. The perceived performance model of testing satisfaction developed by scholars Tse and Wilton shows that user satisfaction can be evaluated by measuring the actual perception of product performance [87]. When the perception exceeds expectations, tourists will have a satisfactory psychological state. High levels of perceived value can stimulate tourists' positive emotional responses, thereby enhancing satisfaction [88]. There is a significant positive relationship between customer perceived value and satisfaction [89]. Moreover, Wang et al. [90] found that tourist perceived value is a prerequisite variable for tourists' satisfaction and that there is a positive correlation between tourists' satisfaction and tourist perceived value. Based on the discussion above, this study advances the following hypothesis:

**H2.** *The perceived value of tourists' experience of STT has a significant impact on tourists' satisfaction in visitor attractions.*

3.1.3. Tourists' Satisfaction and Behavioral Intention

Satisfaction or dissatisfaction in a tourism context and setting will generate a positive or negative emotional response; higher satisfaction may positively impact behavioral intention. Cardozo proposed a customer satisfaction theory in the marketing literature [83]. In a study on coastal tourism destinations, Pizam first introduced satisfaction theory in the tourism field [49]. The relationship between tourist satisfaction and post-consumption behavior was gradually explored [84]. After studying international tourists in Cyprus, Yoon found that tourists with high satisfaction are more willing to revisit and recommend [70]. Based on the theory of "cognition–emotion–intention," Sun demonstrated that tourists'

satisfaction directly affects behavioral intention [85]. Baker and Bigné [86] equally endorsed the significantly positive influence of satisfaction on willingness to pay a premium; the higher the tourists' delight, the higher the willingness to pay a premium. Based on this discussion, this study postulates the next three hypotheses:

**H3.** *Tourists' satisfaction significantly impacts the willingness to recommend visitor attractions.*

**H4.** *Tourists' satisfaction significantly impacts the intention to revisit visitor attractions.*

**H5.** *Tourists' satisfaction significantly impacts the willingness to pay a premium in visitor attractions.*

### 3.2. Research Model

　　The suggested conceptual/research model has explored the relationship between the attributes of STTs, the tourists' perceived value of and satisfaction generated from the smart technology-enhanced experience, and their post-consumption behavioral intention. The latter is assessed/measured by three criteria/variables: intention to revisit, willingness to recommend, and willingness to pay a premium. Figure 1 depicts the suggested model.

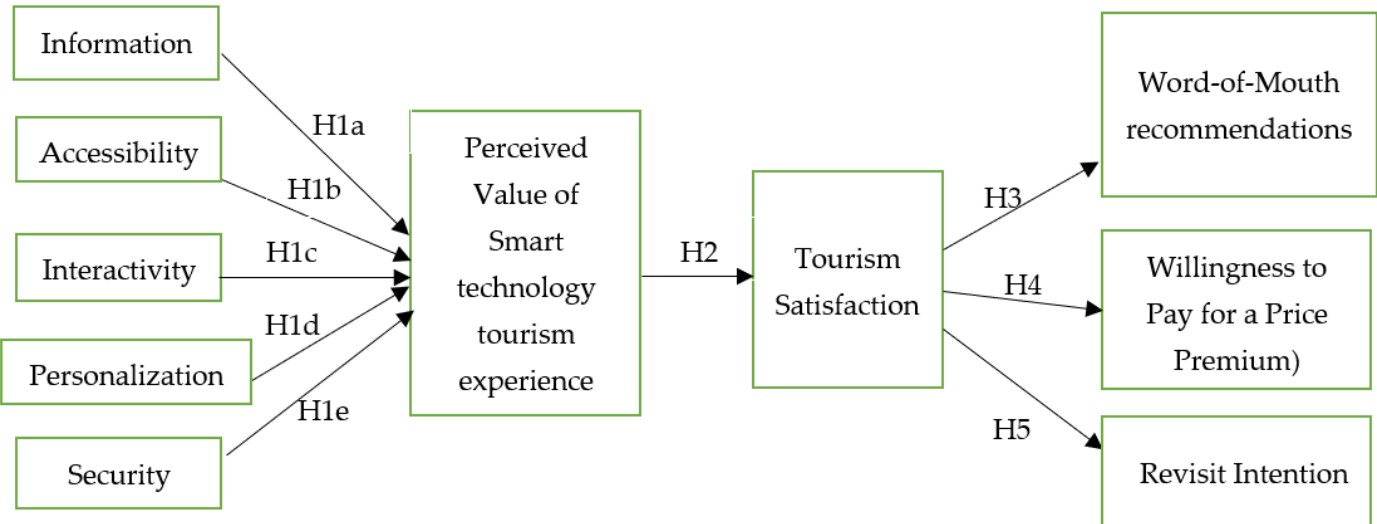

**Figure 1.** Research model.

　　All research constructs (Figure 1) were adapted and modified from previous studies. The constructs were measured using multi-measurement items adapted from the extant literature. The experience of STTs was measured by five variables: information, accessibility, interactivity, personalization, and security. A three-item scale was used for each dimension. Perceived value included five measurement items and tourist satisfaction included four. The construct of behavioral intentions was evaluated using three variables, namely revisit intention (four items), word-of-mouth recommendations (four items), and WPPP (three items). The measurement items for each variable are shown in Table 1. For all forty items, a seven-point Likert scale was used.

**Table 1.** Research constructs and measurement scale.

| Variable | Measurement Items | Supporting Studies |
|---|---|---|
| Information | INF1: Information provided about my travel via smart tourism technologies is useful/helpful.<br>INF2: Smart tourism technologies enable me to complete my travels with reliable and detailed information.<br>INF3: Smart tourism technologies contribute to minimizing my travel concerns. | No & Kim. (2015) [30]<br>Lee et al. (2018) [41]<br>Yoo al. (2017) [91] |
| Accessibility | ACC1: I can use smart tourism technologies anywhere and at any time during my travels.<br>ACC2: Smart tourism technologies are easily accessible during my travels.<br>ACC3: Smart tourism technologies are easily found without complicated processes when traveling. | No & Kim. (2015) [30]<br>Lee et al. (2018) [41] |
| Interactivity | INT1: Smart tourism technologies are interactive when I am traveling.<br>INT2: Smart tourism technologies are highly responsive during my travels.<br>INT3: It is easy to share information and content on smart tourism technologies during my travels. | No & Kim. (2015) [30]<br>Lee et al. (2018) [41]<br>Yoo al. (2017) [91] |
| Personalization | PER1: I received customized/tailored information on smart tourism technologies when I was traveling.<br>PER2: Smart tourism technologies provide me with easy-to-follow links and tips while traveling.<br>PER3: I can get personalized information through interactions with smart tourism technologies while traveling. | No & Kim. (2015) [30]<br>Lee et al. (2018) [41] |
| Security | SEC1: Smart tourism technologies protect my personal and sensitive information.<br>SEC2: Smart tourism technologies respect my privacy and the safety of my transactions.<br>SEC3: Smart tourism technologies are trustworthy and reliable. | Huang et al. (2017) [19]<br>No & Kim. (2015) [30]<br>Zeithaml et al. (1996) [92] |
| Perceived Value of Smart Technology Tourism Experience | PV1: Considering the price I paid, it is worth using smart tourism technologies.<br>PV2: Considering the time and effort devoted to them, it is worth using smart tourism technologies.<br>PV3: The overall value of using smart tourism technologies is high; high value for money.<br>PV4: I have a very good feeling about my experiences with smart tourism technologies.<br>PV5: The use of smart travel technologies is pleasant and entertaining/fun. | Lee et al. (2018) [41]<br>Sweeney & G N. (2001) [93]<br>Petrick J F. (2002) [94]<br>Lee & Yoon & Lee. (2007) [95] |
| Tourists' Satisfaction (SAT) | SAT1: I am happy with the STT experience at this visitor attraction.<br>SAT2. I really enjoy using STTs.<br>SAT3. I am delighted to use STT at this visitor attraction.<br>SAT4: I am satisfied with the experience service quality provided by STTs. | Lee et al. (2018) [41]<br>Yoon & Uysal. (2005) [76]<br>Oliver (1997) [96]<br>Neal et al. (1999) [97]<br>Bigne et al. (2001) [98]<br>Kim et al. (2015) [99] |
| Word-of-Mouth (WOM) Recommendations | WOM1: I would recommend STTs to my family, friends, and peers.WOM2: I will tell my family about my positive experiences with STTs.<br>WOM3: I will speak highly of (say positive things about) STTs.<br>WOM4: I will post positive reviews and comments about STTs on social media. | Yoon & Uysal. (2005) [76]<br>Zeithaml et al. 1996 [92]<br>Bigne et al. (2001) [98]<br>Cronin et al. (2000) [100] |

**Table 1.** *Cont.*

| Variable | Measurement Items | Supporting Studies |
|---|---|---|
| Revisit Intention (RIN) | RIN1: I want to experience STTs again in the future.<br>RIN2: I would like to use STTs again in visitor attractions or other tourism settings.<br>RIN3: I plan to visit attractions with STTs again in the future.<br>RIN4: If I visit a visitor attraction again, one of my main motivations is to use STTs again. | Bigne et al. (2001) [98]<br>Kim et al. (2015) [99]<br>Jang & Feng (2007) [101]<br>Kim et al. (2010) [102]<br>Hung et al. (2016) [103]<br>Zhang et al. (2016) [104] |
| Willingness to Pay a Price Premium (WPP) | WPP1: I am willing to pay a premium for STTs in general.<br>WPP2:I can accept a price increase for intelligent tourism technology.<br>WPP3:I am willing to pay a higher price for visitor attractions offering smart tourism infrastructure. | Zeithaml et al. 1996 [92]<br>Zhang &Bloemer.(2008) [105]<br>Biswas & Roy.(2015) [106]<br>Zhang et al. (2020) [107] |

## 4. Empirical Study: Research Design and Methodology

### 4.1. Study Site

Liangzhu Museum is located in Yuhang District, Hangzhou City, Zhejiang Province, China. This museum covers an area of 40,000 square meters, with an exhibition area of more than 4000 m$^2$. There are three permanent and one temporary exhibition halls. More than 600 precious cultural relics of the Liangzhu Culture period, such as jade, stone, pottery, lacquer, and wood, are displayed in the courtyard. It displays the archaeological achievements and heritage value of the Liangzhu Site and Liangzhu Culture and reflects the unique contribution of the Liangzhu Civilization [108].

Liangzhu Museum completed a transformation and upgrading of its infrastructure and reopened in 2018. It combines traditional museum displays and digital exhibitions through virtual reality, augmented reality, 5G technology, artificial intelligence, and 3D printing. It utilizes hi-tech means such as visual recognition and speech recognition. It provides tourists with tourism experience services such as scene restoration, AR map navigation, digital sandbox, virtual dragoman, smart information retrieval, BBS, and mobile payment. A visiting experience immersion was also implemented.

The Liangzhu Museum was chosen as the study attraction for three reasons, namely: (i) It has been included on the World Heritage List since 2019. (ii) Liangzhu Museum has become one of the representatives of China Smart Museums. Compared with other attractions, the implementation of STTs in the Liangzhu Museum is more integrated and diverse. (iii) In 2020, Liangzhu Museum received more than one million visitors and was named "2020 Top Ten Influence Museum".

### 4.2. Instrument Development

Due to the impact of the COVID-19 pandemic, inter-provincial tourism has been suspended in many regions of China. The research team abandoned the initial idea of all offline/on-site surveys (face-to-face interviews) and opted for a combination of online and offline questionnaires (see Survey S1). The initial plan was a volume of 500 questionnaires, 100 offline, and 400 online. Tourists who visited Liangzhu Museum at least once were selected as respondents (convenience sampling). The data were collected in December 2021. Following the screening, 34 questionnaires were not taken into account as they were incomplete. Hence, 486 valid questionnaires were obtained (a return rate of 93.5%).

### 4.3. Data Collection and Analysis

4.3.1. Sample Characteristics

The demographic and behavioral characteristics of the sample are depicted in Table 2. The sample consisted mainly of females (305 or 62.8%) from the age group of 18–25 (229 or 47.1%). Most respondents were university educated (65%) and lived in cities and towns (89%). Most respondents did one or two visits and thought that the Liangzhu Museum is important for cultural tourism.

**Table 2.** Sample population characteristics (n = 486).

| Characteristics | Frequency (n) | Percentage (%) |
|---|---|---|
| Gender | | |
| Male | 181 | 37.2 |
| Female | 305 | 62.8 |
| Age group | | |
| 18–25 | 229 | 47.1 |
| 26–30 | 104 | 21.4 |
| 31–40 | 87 | 17.9 |
| 41–50 | 48 | 9.9 |
| 51–60 | 11 | 2.3 |
| 60+ | 7 | 1.4 |
| Area of permanent | | |
| City | 278 | 57.2 |
| County | 156 | 32.1 |
| Town | 52 | 10.7 |
| Educational level | | |
| Junior school and below | 12 | 2.5 |
| High school or equivalent | 31 | 6.4 |
| University or equivalent | 315 | 64.8 |
| Master and above (postgraduate) | 128 | 26.2 |
| Capacity | | |
| Professional and technical personnel (teachers, doctors, engineers) | 67 | 13.8 |
| Tertiary industry personnel (catering service, driver, salesman, etc.) | 72 | 14.8 |
| Enterprise staff | 87 | 17.9 |
| Public sector employees, civil servants, government staff | 42 | 8.6 |
| Government staff | 126 | 25.9 |
| Freelancer | 42 | 8.6 |
| Worker | 23 | 4.7 |
| Laborer engaged in forestry | 21 | 4.3 |
| Other | 6 | 1.4 |
| Motivations | | |
| Tourism and Leisure | 285 | 58.6 |
| Games | 253 | 52.1 |
| Education | 235 | 48.9 |
| Entertainment | 221 | 45.5 |
| Work | 198 | 40.7 |
| Other reasons | 48 | 9.9 |
| Number of visits to tourist attractions with intelligent tourism technology | | |
| 1–2 | 173 | 35.6 |
| 3–4 | 147 | 30.2 |
| 5–7 | 113 | 23.3 |
| 8+ | 53 | 10.9 |
| Number of visits to case museums | | |
| 1 | 389 | 80.1 |
| 2 | 87 | 17.9 |
| 3+ | 10 | 2.0 |

### 4.3.2. Measurement Model

The average is used to measure the variable value's average level and concentration trend based on a formal research sample. The average score of information, accessibility, interactivity, personalization, and security in the technical attributes of STTs was 5.196, 5.172, 5.215, 5.139, and 5.282, respectively.



Cronbach's α reliability coefficient method was used for reliability analysis. Based on the reference standard proposed by Cronbach [109], when the α coefficient is between 0.6 and 0.8 the reliability is acceptable. When the α coefficient is greater than 0.8, the reliability is preferable. As shown in Table 3, all Cronbach's α coefficients were greater than 0.7, showing good reliability for all items.

**Table 3.** Reliability and validity analysis.

| Variables | Items | | Mean | Standard Deviation | Standard Loading | T-Value | Composite Reliability | AVE | Cronbach's α | Variables |
|---|---|---|---|---|---|---|---|---|---|---|
| Attributes of STTs | Information | INF1 | 5.22 | 1.236 | 0.629 | 60.065 | | 0.792 | 0.563 | 0.781 |
| | | INF2 | 5.17 | 1.234 | 0.910 | 59.507 | | | | |
| | | INF3 | 5.20 | 1.274 | 0.689 | 58.003 | | | | |
| | Accessibility | ACCE1 | 5.53 | 1.321 | 0.794 | 53.014 | | 0.881 | 0.716 | 0.876 |
| | | ACCE2 | 5.50 | 1.360 | 0.926 | 54.650 | | | | |
| | | ACCE3 | 5.45 | 1.396 | 0.808 | 46.509 | | | | |
| | Interactivity | INT1 | 5.19 | 1.237 | 0.772 | 59.676 | | 0.828 | 0.606 | 0.816 |
| | | INT2 | 5.15 | 1.189 | 0.872 | 61.593 | | | | |
| | | INT3 | 5.30 | 1.176 | 0.680 | 63.995 | | | | |
| | Personalization | PRE1 | 5.15 | 1.375 | 0.758 | 53.253 | | 0.822 | 0.609 | 0.821 |
| | | PRE2 | 5.10 | 1.227 | 0.749 | 59.100 | | | | |
| | | PRE3 | 5.16 | 1.333 | 0.831 | 55.017 | | | | |
| | Security | SEC1 | 5.33 | 1.130 | 0.755 | 67.044 | | 0.831 | 0.553 | 0.806 |
| | | SEC2 | 5.23 | 1.142 | 0.809 | 65.132 | | | | |
| | | SEC3 | 5.28 | 1.135 | 0.725 | 66.147 | | | | |
| Perceived Value of Smart Technology Tourism Experience (STTE) | | PV1 | 5.28 | 1.215 | 0.797 | 61.769 | | 0.878 | 0.591 | 0.878 |
| | | PV2 | 5.28 | 1.151 | 0.726 | 65.141 | | | | |
| | | PV3 | 5.31 | 1.170 | 0.790 | 64.487 | | | | |
| | | PV4 | 5.23 | 1.142 | 0.758 | 65.132 | | | | |
| | | PV5 | 5.18 | 1.179 | 0.770 | 62.402 | | | | |
| Tourism Satisfaction | | SAT1 | 5.29 | 1.088 | 0.667 | 69.155 | | 0.821 | 0.536 | 0.820 |
| | | SAT2 | 5.32 | 1.093 | 0.788 | 69.207 | | | | |
| | | SAT3 | 5.36 | 1.103 | 0.777 | 69.049 | | | | |
| | | SAT4 | 5.29 | 1.096 | 0.689 | 68.576 | | | | |
| Word-of-Mouth (WOM) Recommendations | | WOM1 | 5.28 | 1.174 | 0.672 | 63.961 | | 0.830 | 0.552 | 0.827 |
| | | WOM2 | 5.28 | 1.203 | 0.782 | 62.403 | | | | |
| | | WOM3 | 5.21 | 1.193 | 0.813 | 62.128 | | | | |
| | | WOM4 | 5.13 | 1.279 | 0.694 | 57.005 | | | | |
| Revisit Intention | | RIN1 | 5.30 | 1.215 | 0.713 | 62.136 | | 0.816 | 0.528 | 0.812 |
| | | RIN2 | 5.19 | 1.175 | 0.805 | 63.346 | | | | |
| | | RIN3 | 5.32 | 1.196 | 0.742 | 63.514 | | | | |
| | | RIN4 | 5.18 | 1.304 | 0.635 | 56.960 | | | | |
| Willingness to Pay for a Price Premium (WPP) | | WPP1 | 4.77 | 1.438 | 0.804 | 47.174 | | 0.869 | 0.691 | 0.859 |
| | | WPP2 | 4.95 | 1.408 | 0.741 | 49.926 | | | | |
| | | WPP3 | 4.65 | 1.499 | 0.937 | 45.254 | | | | |

Validity analysis was conducted for both convergent validity (CV) and discriminant validity (DV) to reflect the authenticity and validity of the questionnaire data [110]. The CV test included composite reliability (CR) and average variance extracted (AVE). According to Table 3, the factor load was higher than the standard of 0.5 [111]. The CR was higher than the standard of 0.6, indicating that each variable has good CR [112]. The AVE was higher than the standard of 0.5. The DV is shown in Table 4. The results show a remarkable correlation between constructs ($p < 0.01$). In addition, the absolute value of the correlation coefficient was less than 0.5 and less than the square root of the corresponding AVE [113]. That is to say, there is a certain correlation between each latent variable, and there is a certain degree of discrimination between them, indicating that the DV of the scale data is ideal [114].

**Table 4.** Discriminant validity.

| | 1 | 2 | 3 | 4 | 5 | 6 | 7 | 8 | 9 |
|---|---|---|---|---|---|---|---|---|---|
| 1. INF | 0.560 | | | | | | | | |
| 2. ACCE | 0.361 | 0.632 | | | | | | | |
| 3. INT | 0.070 | 0.238 | 0.580 | | | | | | |
| 4. PER | 0.335 | 0.188 | 0.269 | 0.571 | | | | | |
| 5. SEC | 0.151 | 0.144 | 0.242 | 0.120 | 0.670 | | | | |
| 6. SAT | 0.074 | 0.167 | 0.133 | 0.200 | 0.139 | 0.538 | | | |
| 7. WOM | 0.067 | 0.215 | 0.322 | 0.361 | 0.185 | 0.212 | 0.548 | | |
| 8. RIN | 0.201 | 0.273 | 0.208 | 0.301 | 0.234 | 0.275 | 0.245 | 0.540 | |
| 9. WPP | 0.213 | 0.095 | 0.116 | 0.050 | 0.252 | 0.154 | 0.136 | 0.166 | 0.686 |
| Square root of AVE | 0.748 | 0.795 | 0.762 | 0.756 | 0.816 | 0.733 | 0.740 | 0.735 | 0.828 |

INF: information, ACCE: accessibility, INT: interactivity, PER: personalization, SEC: security, SAT: tourism satisfaction, WOM: word-of-mouth (WOM) recommendations, RIN: revisit intention, WPP: willingness to pay a price premium.

## 5. Test of Structural Model and Hypotheses

This study used a regression model to show the relationship between variables. The validity of the model is evaluated by checking for multicollinearity. As shown in Table 5, the variance inflation factor values range from 1.164 to 2.712. All values of VIF are acceptable and thus there is no collinearity among the variables. Additionally, the coefficient of determination was measured, as shown in Figure 2. Tourism satisfaction explained 58.8% of the variance, word-of-mouth recommendations explained 54.1% of the variance, revisit intention explained 61.3% of the variance, and willingness to pay a price premium explained 51.6% of the variance.

**Table 5.** Multicollinearity.

| Variables | VIF |
|---|---|
| (1) Information | 2.518 |
| (2) Accessibility | 1.164 |
| (3) Interactivity | 1.203 |
| (4) Personalization | 2.712 |
| (5) Security | 1.503 |

Normally, the standard path coefficient represents the influence relationship between variables. If it is significant, it indicates an obvious effect between variables. As shown in Table 6 and Figure 2, this study has ten variables. Five first-order variables are used to create a second-order variable—perceived value of smart technology tourism experience. Firstly, this study analyzed whether these five first-order variables are related to

second-order variables. The results showed a significant correlation between attributes (the accessibility having the strongest aboriginality) of STTs and tourists' perceptions of their smart technology tourism experience. Therefore, the results support Hypothesis 1.

The path coefficient of perceived value affecting tourist satisfaction is 0.329. The path showed a 0.01 ($p < 0.001$) level of aboriginality, indicating that tourists' perceptions of their smart technology tourism experiences significantly positively impact tourist satisfaction. The results showed that tourists' perception of STTs positively correlates with tourist satisfaction and thus the test results support Hypothesis 2. Regarding the hypothesis that tourists' satisfaction affects tourists' behavior intentions, the standardized path coefficient of tourists' satisfaction to WOM recommendations was 0.307, showing a significance level of 0.01 ($p < 0.001$). The standardized path coefficient of tourist satisfaction to revisit intention was 0.336, showing a significance level of 0.01 ($p < 0.001$). The standardized path coefficient of tourist satisfaction to the willingness to pay a price premium (WPP) was 0.160, indicating a significance level of 0.05 ($p = 0.021 < 0.05$). Hence, the tourists' satisfaction significantly impacts WOM recommendations, revisit intention, and WPP. Therefore, the results support Hypotheses 3, 4, and 5. In summary, the model supports all the research hypotheses.

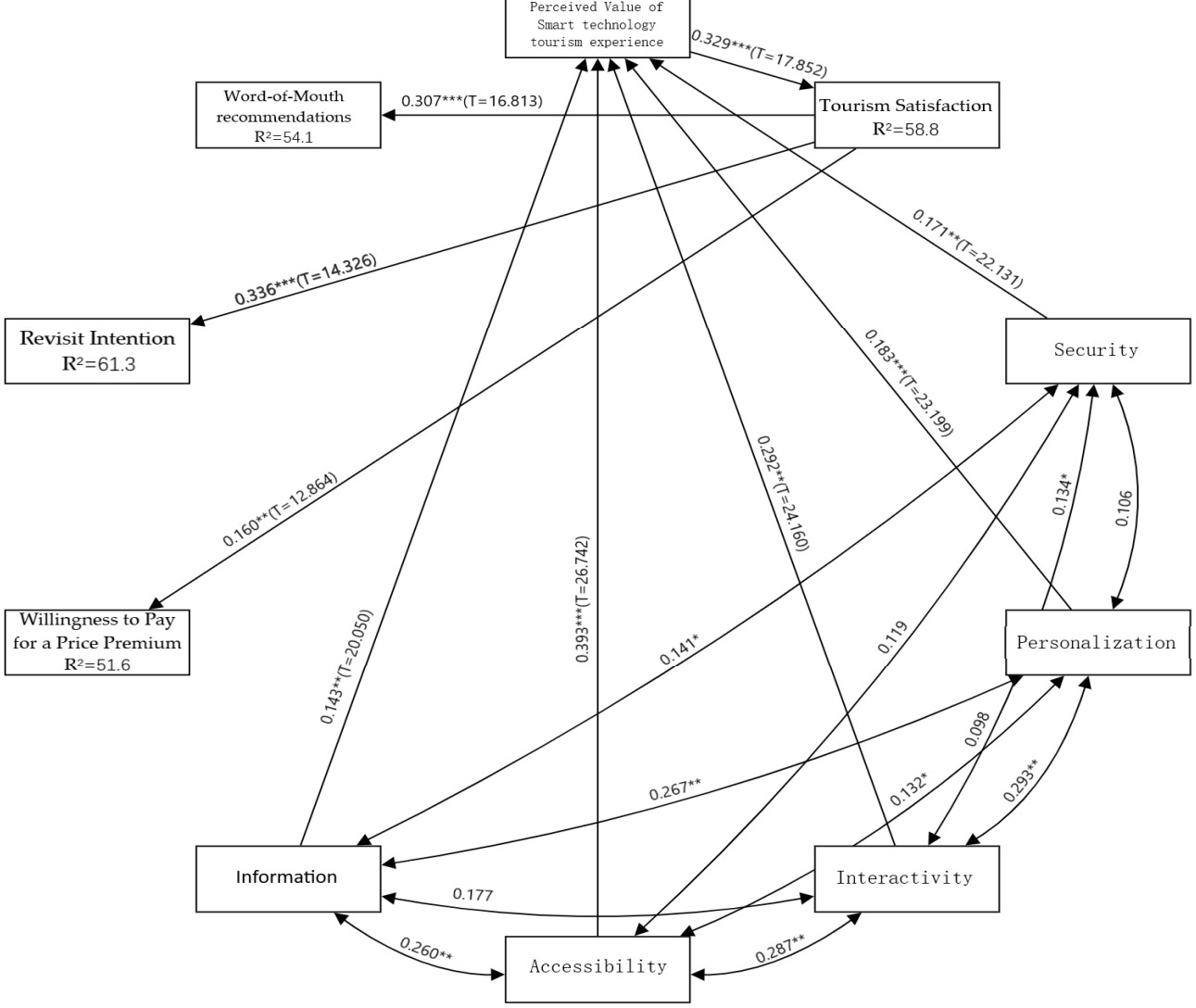

**Figure 2.** Structural model results. ***: $p < 0.001$; **: $p < 0.01$; *: $p < 0.05$.

**Table 6.** Path Analysis and Hypothesis Validation.

| | Hypothesis | Non-Standardized Path Coefficient | Standardized Path Coefficients | z | T | *p*-Value | Result |
|---|---|---|---|---|---|---|---|
| H1a | Information → STATE | 0.131 | 0.143 | 2.111 | 20.050 | 0.035 | supported |
| H1b | Accessibility → STATE | 0.502 | 0.393 | 3.189 | 26.742 | 0.001 | supported |
| H1c | Interactivity → STATE | 0.246 | 0.292 | 1.759 | 24.160 | 0.017 | supported |
| H1d | Personalization → STATE | 0.183 | 0.183 | 2.701 | 23.199 | 0.007 | supported |
| H1e | Security → STATE | 0.149 | 0.171 | 1.683 | 22.131 | 0.029 | supported |
| H2 | STTS → Tourism Satisfaction | 0.302 | 0.329 | 4.949 | 17.852 | 0.000 | supported |
| H3 | Tourism Satisfaction → Word-of-Mouth Recommendations | 0.343 | 0.307 | 4.593 | 16.813 | 0.000 | supported |
| H4 | Tourism Satisfaction → Willingness to Pay for a Price Premium | 0.250 | 0.160 | 2.308 | 12.864 | 0.021 | supported |
| H5 | Tourism Satisfaction→Revisit Intention | 0.378 | 0.336 | 5.077 | 14.326 | 0.000 | supported |

## 6. Discussion and Conclusions

The study's main purpose was to investigate the influence of the attributes of STTs on tourists' experiences enhanced by STTs. This research capitalized on the extant literature—conceptualization of dimensions of STTs—and elaborated on a comprehensive model of the attributes of STTs, tourists' satisfaction, and tourists' behavioral intention. The STTs attributes theory proposed by Huang was adopted, extended by the security dimension [19]. Tourists' perception about the five attributes of STTs (information, accessibility, interactivity, personalization, security) impacts the tourists' satisfaction through the perceived value, influencing the tourists' consumption behavior after completing the visit experience.

Of the five attributes of STTs, accessibility was found to be the strongest predictor. According to Emiliani, "the consideration of accessibility should be shifted from post-design stage to pre-design stage" (p. 256) [115]. This involves "putting users and potential users at the core of technology development, users" demands are the focus of design (p. 257) [116]. Our study's findings are consistent with those of previous studies. Tourists can use STTs anywhere and at any time in the tourism decision-making process without complex processes or significant effort so they have a strong perception of easiness. Thus, tourists can spend more time enjoying tourism activities based on STTs. Only STTs adapted to tourists' requirements can improve tourists' perception of the accessibility of STTs.

Second, interactivity proved to be the second strongest predictor. Based on previous studies, interactivity is operationally composed of three principal elements, i.e., properties of technology, attributes of communication contexts, and user perceptions. Concerning individuals, it also refers to users' ability to perceive the experience as being a simulation of interpersonal communication [117]. Therefore, high-level interaction can motivate tourists to use STTs more actively. Visitors to an attraction actively search for tourism-related information through STTs. Through the same channels, staff also collect tourists' preferences, the aim being to provide customized products and services to meet their needs [33].

Contrary to expectations, this study does not endorse the significance of the attribute of information. Tourism is a product of informatization, so high-quality information is crucial [30]. Information quality can be described as the degree of fit between information and needs, namely whether the information provided can meet one's needs and improve the experience quality [118]. It could be argued that museums, as visitor attractions, are special/particular. Although their visitation and influence are increasing, museums' educational function is still far beyond leisure and recreation in most Chinese tourists' minds. Only a small portion of visitors share reviews on tourism websites about their visit experience at the Liangzhu Museum. The information available on the official website lacks empathy and does not contribute to overcoming visitors' concerns. Overall, the significance of STTs in order of importance is accessibility, interactivity, personalization, security, and information. It is apparent that the first two dimensions—accessibility and interactivity— play an important role in improving the smart technology-enhanced tourism experience.

Therefore, visitor attractions and museums should start from the demand/visitor side and constantly optimize the use of STT infrastructure and services, simplify the usage of STTs, and strengthen the contact and communication between tourists and other stakeholders to improve the accessibility and interactivity of STTs, thereby further enhancing the tourists' perception of the usability and usefulness of the STTs. At the same time, attractions should pay attention to communications and promotional mixes. The style is serious and formal, and there is less post-tourist information feedback with tourists' subjective color on relevant tourism information websites. Therefore, visitor attractions can take the form of small gifts to solicit user-generated content and positive reviews on tourism and social networking sites, thereby mitigating visitors' concerns. This is regarded as an efficient communication strategy. Finally, software and services designers should pay more attention to customized service performance and plan more diversified experiences when developing related STTs (e.g., platforms). Likewise, they have to carefully consider the security and privacy protection, enhancing the dimensions of personalization and security attributes to make STTs more responsive to tourists' requirements.

It is believed that satisfaction is a more emotional response to perceived experience, and perceived experience is a powerful predictor of tourist satisfaction [119]. Previous studies revealed a correlation between service perception experience and satisfaction [120]. The first hypothesis (about the positive relationship between the perceived value of STTs and tourists' satisfaction) was confirmed, in line with the extant literature [121,122]. When tourists highly value their STT-supported experience, they will have positive emotions and thus produce high tourist satisfaction. Visitors' high perception of STTs can lead to higher satisfaction, which agrees with scholars such as Santiago who indicate that the high perception of the experience is a driving force for improving tourist satisfaction [123]. Therefore, it is suggested that visitor attractions' managers and marketers should design, manage and operate appropriate STT infrastructure and services based on very high quality intelligence. In China, the infrastructure and management of smart museums are still at the initial introduction stage. Museums and other visitor attractions that have undergone a 'smart' transformation should constantly strengthen STT infrastructure and collect online data about tourists' opinions to improve tourists' satisfaction with efficient facilities and services.

The significant and positive relationship between satisfaction and revisit intention and WOM recommendations are also endorsed by the literature [124] and supported by empirical studies in the context of suburban tourism [125] and wine tourism [126]. Likewise, Baker [80] and Bigné [127] demonstrated significant and positive impacts of consumers' satisfaction on WPP, a suggestion supported by Xu [128] in the context of rural tourism. Our study's findings confirmed that visitors' satisfaction positively impacts all three criteria investigated, i.e., revisit intention, WOM recommendations, and WPP.

The above outlined findings of our study have theoretical and practical implications. From an academic perspective, this study suggested and validated a framework for exploring the technology-enhanced tourism experience. This framework provides researchers

with an integrative approach to consider and investigate the features/attributes of this kind of experience. The suggested approach allows us to establish a relationship between STTs' attributes and tourist consumer behavior and consequently to acquire a better understanding of the value and utility of STTs within specific tourism contexts and settings. The proposed conceptual model should be useful in future research endeavors in this field.

This knowledge of the actual content and operational implementation of STTs in the context of visitor attractions (e.g., museum) offers insights and valuable input to help enhance the tourism experience and increase visitor satisfaction. The ultimate strategic aim is the improvement of their positioning in a highly competitive market.

Therefore, the following recommendations for the visitor attractions industry could be formulated. First, attach more importance to the attractiveness and appeal of STTs. STTs should support the exhibitions and exhibits. These digital technologies are not the main content of museum/attraction experiences; managers and marketers should regard them as supporting tools contributing to improving the visitor experience and the museum's efficient management and smooth operation. Second, there is a need to enrich STTs. STTs in Chinese museums provide fewer entertainment opportunities. Museums should use STTs to create more enjoyable experience opportunities to enhance the participation and interactive experience. Third, there is also a need for integrated promotion and communication with visitors to understand and take full advantage of the added value of STTs. Due to the high investment in the infrastructure and services of STTs, the operational expenses are high and attractions cannot charge their visitors accordingly. The only way/strategy to do so is to convince their potential visitors about the added value and the value for money of all facilities and services supported by STTs. This is the real meaning of a digital technology-supported visit/tourism experience.

## 7. Limitations and Future Research Directions

The first limitation is the data collection and survey technique. They were affected by the COVID-19 pandemic, resulting in less on-site data collection. The second limitation is the sampling method. Since the empirical survey was conducted in Hangzhou City, Zhejiang Province, China, the offline respondents were mainly local permanent residents. Therefore, the convenience sampling is an issue. Third, the study focused on five dimensions/attributes—information, accessibility, interactivity, personalization, and security—to evaluate the visitors' perception of the STT-enhanced experience, drawing on the attribute theory of STTs proposed by Huang [4] and Kim [27]. Other dimensions may need to be further studied in the future.

Moreover, future research projects could explore additional constructs of post-experience behavior regarding the conceptual model. A fourth limitation is the study's context—a Chinese museum. Researchers could examine the same topic (smart technology-supported experiences) in other visitor attractions and other countries in Asia or Europe.

**Supplementary Materials:** The following supporting information can be downloaded at: https://www.mdpi.com/article/10.3390/su14053048/s1, Survey S1: Questionnaire: Research on the Impact of Smart Tourism Technologies (STTs) on Tourists' Experience.

**Author Contributions:** Conceptualization, Y.Z. and M.S.; methodology, Y.Z. and M.S.; software, Y.Z.; validation, Y.Z., M.S., and S.S.; formal analysis, Y.Z., M.S., and S.S.; investigation, Y.Z. and S.S.; resources, S.S.; data curation, Y.Z.; writing—original draft preparation: Y.Z.; writing—review and editing: Y.Z., M.S., and S.S.; project administration and supervision: S.S. All authors have read and agreed to the published version of the manuscript.

**Funding:** This research received no external funding.

**Institutional Review Board Statement:** Ethical review and approval were waived for this study/research due to the following reasons: (1) Our research project was on humans; however, it did not include any private sensitive or personal information about the participants in the exploratory study; (2) Ethics committee or institutional review board approval is not requested by Ningbo University (NBU) for research projects conducted by postgraduate students; (3) this study has been carried out within the framework of the first authors' postgraduate studies and under the professors' (second and third authors) supervision and close monitoring. All postgraduate students fully comply with the ethical regulations of NBU.

**Informed Consent Statement:** Informed consent was obtained from all subjects involved in the study.

**Data Availability Statement:** The data that support the findings of this study are openly available in [repository name "figshare"] at http://doi.org/10.6084/m9.figshare.19236729 (accessed on 1 September 2021).

**Conflicts of Interest:** The authors declare no conflict of interest.

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
