# Peer review of "Investigating the Impact of Smart Tourism Technologies on Tourists’ Experiences"

_sustainability, doi:10.3390/su14053048_

Round 1

Reviewer 1 Report

Dear authors,

First of all, congratulations for your work. However, manuscript should be modified following considerations described as follows:

Lines 8-19 (abstract): please include methodology used explicitly (structural model, regression) in abstract.

Line 17: please do not use acronyms in abstract. Turn STT as Smart tourism technologies.

Line 33: please state complete expression every first time an acronym is used, as VR, in this case.

Line 49: same as before. Please state complete expression every first time an acronym is used, as STT, in this case.

Line 57, and 64-65: please avoid paragraphs with one or two lines. Please rewrite and group it in larger paragraphs instead.

Line 62, 109, 167, 270: be careful with the blank spaces in tourist ‘. There should not exist blank space between tourist and ‘.

Lines 71-76: please name and include the methodology used.

Line 85, 243. Please do not use acronyms in titles.

Lines 236-240. Could be WPPP instead? As “Willingness to Pay a Price Premium”, has three Ps? By the way, I try to search ref. 80 to check it but I cannot found it.

Line 363. Table 2. Why educational level has not horizontal lines? By the way, the Table 2 design is not consistent with the rest of the tables.

Lines 390-414. Please add coefficient of determination (R2) reached in regression model and please state variance inflation factor (VIF) of every variable to discard multicollinearity.

Line 516. Please remove hyphen in “col-lection”.

Line 518. Same in “per-manent”.

Line 519. In what means sample cannot be considered representative? Be careful, as this argument can invalidate all your work. Please rewrite sentence explaining better how your study is consistent.

Line 528. Same in “Eu-rope”.

Author Response

Thank you very much for your comments! Please see attached the responses.

Reviewer 2 Report

The article is written in a very precise and detailed way. The only fundamental comment I have is related to the understanding of what is smart tourism.

https://www.mdpi.com/2071-1050/13/18/10279

I suggest that the authors check and potentially include the above article, and argue why their analyzed innovations contribute to the understanding (and not smartwashing) of the smart tourism field.

Author Response

(The authors gave the same response as above.)

Reviewer 3 Report

This paper aims to investigate the impact of smart tourism technologies (STTs) on tourists’ experience. I consider this paper was fairly well done but could have pushed harder in terms of outlining its novelty and contribution to the field of tourism studies with STTs. The research is interesting, but the main issue is that the contribution to existing STTs theories seems unclear. This limits the value of this paper to a certain extent. While I am in favor of publication, the paper requires more work before it could be considered for publication. This paper does not provide clear statements on STTs to the museum. Most survey items are general questions about STTs not related to museum.

In Table 2, the age group shows that 18- is 26 and 18-25 is 203. What does that mean here? 18-? and 18-25 what’s the difference?

The path coefficient of perceived value affecting tourist satisfaction is 0.329. The path 400 showed a 0.01 (P = 0.000 < 0.01) – Please check writing style of p-value.   

Author Response

(The authors gave the same response as above.)

Round 2

Reviewer 2 Report

Thank you, good work, I don't have additional comments.